# The CD153 vaccine is a senotherapeutic option for preventing the accumulation of senescent T cells in mice

Shota Yoshida [1,2], Hironori Nakagami [2✉], Hiroki Hayashi [2], Yuka Ikeda[3], Jiao Sun [2], Akiko Tenma[3], Hideki Tomioka [3], Tomohiro Kawano[2,4], Munehisa Shimamura [2], Ryuichi Morishita [5] & Hiromi Rakugi [1]

Senotherapy targeting for senescent cells is designed to attenuate age-related dysfunction. Senescent T cells, defined as CD4+ CD44high CD62Llow PD-1+ CD153+ cells, accumulate in visceral adipose tissues (VAT) in obese individuals. Here, we show the long-lasting effect of using CD153 vaccination to remove senescent T cells from high-fat diet (HFD)-induced obese C57BL/6J mice. We administered a CD153 peptide-KLH (keyhole limpet hemocyanin) conjugate vaccine with Alhydrogel (CD153-Alum) or CpG oligodeoxynucleotide (ODN) 1585 (CD153-CpG) and confirmed an increase in anti-CD153 antibody levels that was sustained for several months. After being fed a HFD for 10–11 weeks, adipose senescent T cell accumulation was significantly reduced in the VAT of CD153-CpG-vaccinated mice, accompanied by glucose tolerance and insulin resistance. A complement-dependent cytotoxicity (CDC) assay indicated that the mouse IgG2 antibody produced in the CD153-CpG-vaccinated mice successfully reduced the number of senescent T cells. The CD153-CpG vaccine is an optional tool for senolytic therapy.

[1] Department of Geriatric and General Medicine, Osaka University Graduate School of Medicine, Osaka, Japan. [2] Department of Health Development and Medicine, Osaka University Graduate School of Medicine, Osaka, Japan. [3] R&D, FunPep Co., Ltd., Osaka, Japan. [4] Department of Neurology, Osaka University Graduate School of Medicine, Osaka, Japan. [5] Department of Clinical Gene Therapy, Osaka University Graduate School of Medicine, Osaka, Japan. ✉email: nakagami@gts.med.osaka-u.ac.jp

Senescent cells produce proinflammatory and matrix-degrading molecules, which harm their surrounding non-senescent cells[1–5]. Senotherapy targeting for senescent cells is designed to attenuate age-related dysfunction and promote healthy aging[6–8], and the removal of senescent cells by direct killing, either by apoptotic (senoptosis) or nonapoptotic (senolysis) methods, is an effective serotherapeutic approach. In the genetic model, INK-ATTAC mice, to undergo the inducible elimination of p16$^{Ink4a}$-expressing cells, these mice in which p16$^{Ink4a}$-positive senescent cells were eliminated exhibited a long life span and the attenuation of several aging phenotypes in white adipose tissue, the heart, and the kidney[6]. Senescent cells accumulate in fat in aging, and exercise-mediated reduction as well as genetic clearance improved glucose metabolism or lipotoxicity, respectively[9,10]. Senescent T cells (referred to as senescence-associated T cells; SA-T cells), defined as CD4$^+$ CD44$^{high}$ CD62L$^{low}$ PD-1$^+$ CD153$^+$ cells, accumulate in visceral adipose tissues (VAT) in obese individuals[11] and produce proinflammatory cytokines, causing chronic inflammation, metabolic disorders, and cardiovascular diseases[12,13]. However, it is still unknown whether the selective depletion of senescent T cells can attenuate the age-related pathological changes. Here, we show the long-lasting effect of using CD153 vaccination to prevent the accumulation of adipose senescent T cells from high-fat diet (HFD)-induced obese C57BL/6J mice, accompanied by improved glucose tolerance and insulin resistance. A complement-dependent cytotoxicity (CDC) assay indicates that the mouse IgG2 antibody produced in the CD153-CpG-vaccinated mice successfully reduced the number of senescent T cells.

## Results

### Development of CD153 vaccine to reduce the senescent T cells.
The peptide vaccines for mouse CD153 consisted of a short peptide sequence conjugated to KLH as a carrier protein to induce the specific CD153 antibody, and knowing the CD153 epitope information allowed five candidate peptide to be selected (#A, 116–125 aa; #B, 182–189 aa; #C, 101–108 aa; #D, 76–85 aa; #E, 234–239 aa; Supplementary Fig. 1A) as antigens. The five peptide vaccines (30 μg of the CD153 peptide per mouse) were conjugated to KLH and coadministered with the Alum adjuvant (Alhydrogel) to 7-week-old male C57BL/6J mice twice at 2-week intervals. Evaluation of antibody production by the enzyme-linked immunosorbent assay (ELISA) showed that the titer against mouse CD153-BSA was initially increased on day 14 in mice immunized with the #A and #C vaccines and successfully increased on day 28 in mice immunized with all five vaccines (Supplementary Fig. 1B). The ELISA performed using recombinant mouse CD153 (rmCD153) showed that the antibody induced by the CD153#D vaccine strongly reacted with rmCD153 (Fig. 1a), and western blot analysis also showed that the antibody induced by the CD153#D vaccine specifically and strongly reacted with rmCD153. The commercial anti-CD153 antibody, used as a positive control, and the CD153#D vaccine-induced antibody recognized 30 and 100 ng of the rmCD153 protein but not the recombinant mouse osteopontin (OPN) protein (rmOPN; used as a negative control). On the other hand, the antibody induced by the KLH control vaccine recognized neither the rmCD153 protein nor the rmOPN protein (Fig. 1b). These results demonstrate that CD153#D vaccine-induced antibodies effectively recognize the rmCD153 protein. Therefore, we selected the CD153#D vaccine as a candidate for further experimental studies. Alum adjuvant can induce a Th2 immune response, and CpG adjuvant can induce a Th1 immune response[14,15]. We thus evaluated serum mouse IgG subclass antibodies (IgG1; Th2 immune response, IgG2b, IgG2c and IgG3; Th1 immune response) against CD153

by ELISA to confirm whether the CD153-CpG vaccination can induce more of a Th1 immune response than the CD153-Alum vaccination. In the IgG subclass analysis, the CD153-CpG vaccination induced more IgG2b, IgG2c, and IgG3 than the CD153-Alum vaccination (Fig. 1c), demonstrating that the CD153-CpG vaccination effectively induces the Th1 immune response.

The number of CD153$^+$ SA-T cells increases gradually and systemically with age[16]. Indeed, more CD153$^+$ SA-T cells were observed in the splenic tissues of male C57BL/6J mice at 20 months of age than at 3 months of age (18.7 ± 0.5% vs. 3.8 ± 0.2%; mean ± SEM; two-tailed unpaired $t$-test; $p < 0.0001$; $n = 3$ in each group). In VAT, CD153$^+$ SA-T cells tended to increase more in elderly mice than in young mice (8.2 ± 1.0% vs. 4.3 ± 1.4%; mean ± SEM; two-tailed unpaired $t$-test; $p = 0.08$; $n = 3$ in each group) (Supplementary Fig. 2 and Fig. 1d). Consistent with a previous report[11], the majority of SA-T cells showed higher expression of senescence-associated β-galactosidase (SA-β-gal), a typical cell senescence marker, and phosphorylated histone H2AX at serine 139 (γ-H2AX), a DNA damage and repair marker, compared with the CD153$^-$ counterpart cells, which is indicative of greater exposure to genostress. These results suggest that the SA-T cells show signatures of cell senescence (Supplementary Fig. 3). The development of CD153$^+$ senescent T cells is dependent on autoreactive B cells, especially spontaneous germinal centers, which are driven by continuous administration of the TLR7 ligand[17,18]. Therefore, we investigated splenic CD153$^+$ senescent T cells under continuous administration of R848, the TLR7 ligand, to estimate the efficacy of CD153-CpG vaccination. In accordance with the protocol established in the experimental R848 administration study (Supplementary Fig. 4A), the CD153-CpG vaccine was administered to 8-week-old male C57BL/6J mice and female C57BL/6N mice three times at 2-week intervals, and R848 was then administered three times per week for 4–6 weeks. The anti-CD153 antibody titer induced by the CD153-CpG vaccine maintained a high level during the administration of R848 (Supplementary Fig. 4B). Flow cytometry analysis showed that the proportion of splenic CD153$^+$ senescent T cells was increased in the R848-treated mice at 18 weeks of age and increased to a lesser extent at 16 weeks of age compared with that in phosphate-buffered saline (PBS) control group mice. Notably, the proportion of splenic CD153$^+$ senescent T cells was significantly decreased in mice immunized with the CD153-CpG vaccine compared with those in R848 control group mice and KLH-CpG-vaccinated mice (Fig. 1e and Supplementary Fig. 5A). Although sustained TLR7 stimulation causes splenomegaly due to chronic immune activation[19], we found, in the female C57BL/6N mice, the spleen weights to be significantly increased by continuous R848 stimulation and decreased for mice immunized with the CD153-CpG vaccine compared with those of R848 control group mice (Supplementary Fig. 5B). These results indicate that CD153-CpG vaccination decreases the number of splenic CD153$^+$ senescent T cells and improves splenomegaly, stimulated by chronic immune activation.

### CD153 vaccine prevents the accumulation of adipose senescent T cells.
We further investigated whether vaccinating HFD-loaded mice with the CD153-CpG vaccine could decrease the number of CD153$^+$ senescent T cells in VAT and improve glucose tolerance and insulin resistance. In accordance with the protocol provided in the experimental HFD loading study (Fig. 2a), either the CD153-Alum vaccine or the CD153-CpG vaccine was administered to 7-week-old male C57BL/6J mice multiple times, and HFD loading was then performed for 10–11 weeks. The anti-CD153 antibody titer induced by both the CD153-Alum vaccine and the CD153-CpG vaccine maintained a high level during HFD loading (Fig. 2b). The mouse body weights were not significantly

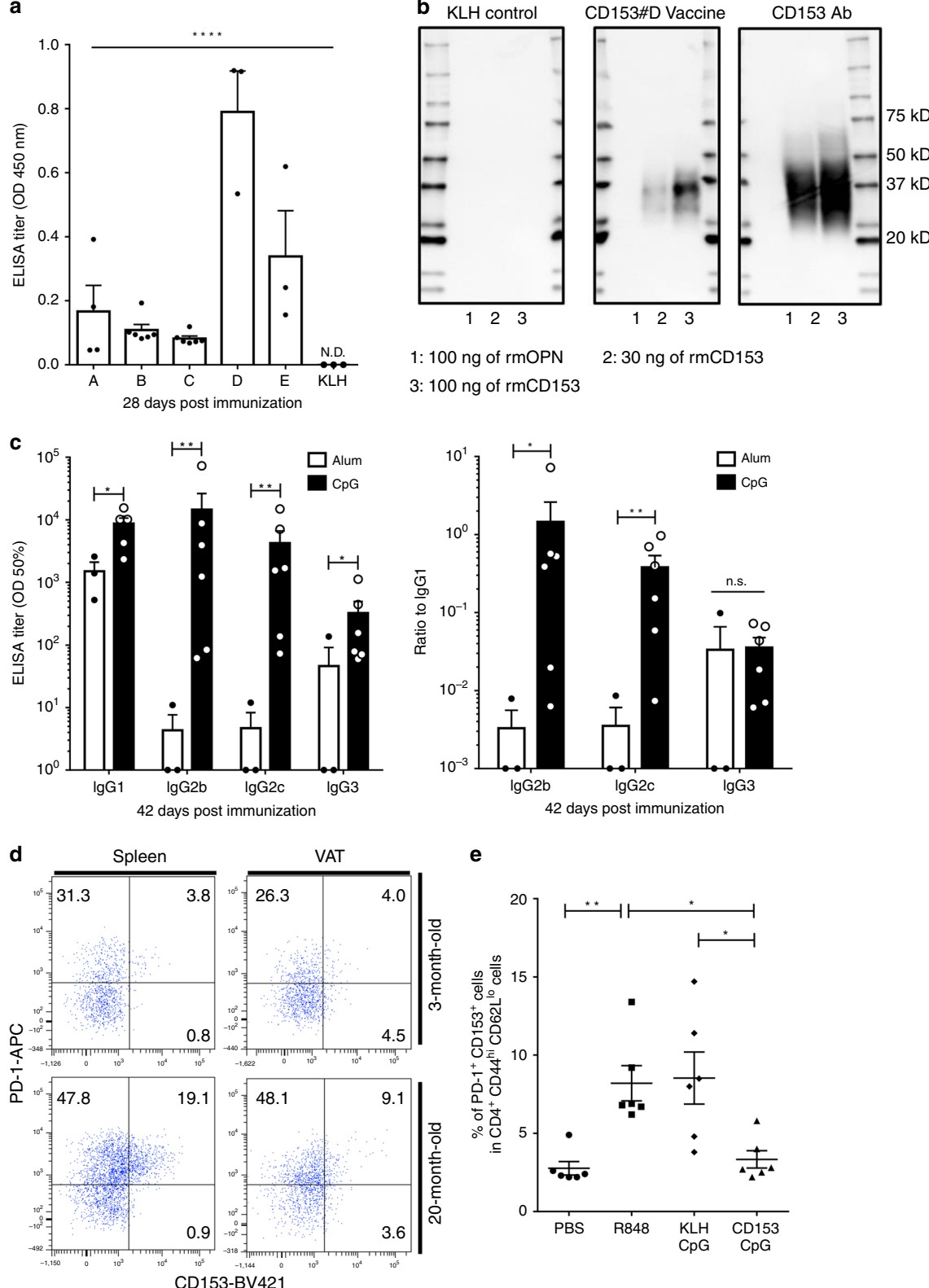

different between the CD153-Alum-vaccinated group and the KLH-Alum-vaccinated group. However, the body weights of mice immunized with the CD153-CpG vaccine were significantly lower than those of mice immunized with the KLH-CpG vaccine and tended to be lower than those of HFD control group mice (Fig. 2c). In addition, the weekly average amount of HFD intake was decreased in mice immunized with the CD153-CpG vaccine

compared with those of the HFD control and KLH-CpG-vaccinated mice (Fig. 2d). Oxygen consumption was estimated as an indirect measurement of metabolism. Mice immunized with the CD153-CpG vaccine exhibited increased oxygen consumption in the dark phase compared with that of mice immunized with the KLH-CpG vaccine, and no significant difference was observed between the CD153-CpG-vaccinated group and the HFD control

**Fig. 1 Development of CD153 vaccine. a–c** Seven-week-old C57BL/6J mice were immunized with the CD153-Alum vaccine (#**a–e**) or the KLH-Alum vaccine on days 0 and 14. **a** The titer against recombinant mouse CD153 is expressed as the OD at 450 nm at 28 days postimmunization. #**a–e**, $n = 4, 6, 6,$ 3, 3, respectively; KLH, $n = 3$. **b** The antibody recognizes the rmCD153 protein by western blot analysis. As a negative control, 100 ng of rmOPN was applied to lane 1, while 30 and 100 ng of rmCD153 were applied to lanes 2 and 3, respectively. **c** The levels of IgG1 (Th2 response) and IgG2b, IgG2c, and IgG3 (Th1 response) in mice immunized with the CD153#D-Alum vaccine ($n = 3$) or the CD153#D-CpG vaccine ($n = 6$) on days 0, 14, and 28. Sera were collected on day 42. IgG1, $p = 0.011$; IgG2b, $p = 0.0067$; IgG2c, $p = 0.0027$; IgG3, $p = 0.0029$. Left; The titer against CD153-BSA is expressed as the half-maximal binding (OD 50%). Right; ratio of the titer against CD153-BSA-specific IgG2b:IgG1 ($p = 0.022$), IgG2c:IgG1 ($p = 0.0091$) and IgG3:IgG1 ($p = 0.21$). **d** The proportions of CD153$^+$ senescent T cells in splenic tissues and VAT of male C57BL/6J mice at the ages of 20 and 3 months (normal diet). Senescent T cells were defined as PD-1$^+$ CD153$^+$ cells in CD4$^+$, CD44$^{hi}$, CD62L$^{lo}$ cells. **e** The proportion of senescent T cells induced by R848 administration in splenic tissues of male C57BL/6J mice ($n = 6$ in each group) at the ages of 18 weeks with or without the CD153-CpG vaccine or the KLH-CpG vaccine. PBS vs. R848, $p = 0.0086$; PBS vs. KLH-CpG, $p = 0.0052$; PBS vs. CD153-CpG, $p = 0.98$; R848 vs. KLH-CpG, $p > 0.99$; R848 vs. CD153-CpG, $p = 0.020$; KLH-CpG vs. CD153-CpG, $p = 0.012$. All the data are expressed as the mean ± SEM. Statistical evaluation was performed by one sided (**a**) and by two sided (**c**, **e**); analysis of variance (ANOVA) test (**a**); unpaired two-tailed t-test (**c**), natural logarithmic transformation; Tukey's multiple comparison test **e**; *$p < 0.05$; **$p < 0.01$; ****$p < 0.0001$; ND, not detected.

group (Fig. 2e). As determined by the intraperitoneal insulin tolerance test (ipITT) and oral glucose tolerance test (OGTT), insulin sensitivity was not different between the CD153-Alum-vaccinated group and the KLH-Alum-vaccinated group (Fig. 3a). In the CD153-CpG-vaccinated group, the insulin sensitivity and area under the curve (AUC) were improved compared with those in the KLH-CpG-vaccinated group but not compared with those in the HFD control group (Fig. 3b, c). Glucose tolerance was not different between the CD153-Alum-vaccinated group and the KLH-Alum-vaccinated group (Fig. 3d). In the CD153-CpG-vaccinated group, the glucose tolerance and AUC were remarkably improved compared with those in the KLH-CpG-vaccinated and HFD control groups (Fig. 3e, f). In addition, the homeostasis model assessment of insulin resistance (HOMA-IR), an index of insulin resistance, was also significantly improved in the CD153-CpG-vaccinated group compared with that in the CD153-KLH-vaccinated group (Fig. 3g). Furthermore, we performed pair-feeding among HFD-loaded mice and administered the CD153-CpG vaccine to normal-diet (ND)-loaded mice to clarify whether our reported outcomes on HFD loading study were truly reflective of the efficacy of CD153-CpG vaccination or were a result of an adverse event of the immunization. Under pair-feeding conditions, the body weight of HFD-loaded mice had almost similar body weight (Supplementary Fig. 6A, B). The body weight and the weekly average amount of ND intake were almost similar between the ND control group and the ND group immunized with the CD153-CpG vaccine (Supplementary Fig. 6A, C). In the CD153-CpG-vaccinated group, despite the similar body weight and food intake, the insulin sensitivity (Supplementary Fig. 7A, B), the glucose tolerance (Supplementary Fig. 7C, D) and the HOMA-IR (Supplementary Fig. 7E) were improved compared with those in the KLH-CpG-vaccinated and HFD control groups. On the other hand, the insulin sensitivity, the glucose tolerance and the HOMA-IR were not significantly different between the ND control group and the ND group immunized with the CD153-CpG vaccine (Supplementary Fig. 7A–E). These results suggest that CD153-CpG vaccination improves obesity-induced metabolic disorders, such as glucose tolerance and insulin resistance, and does not have severe adverse events such as weight loss and loss of appetite.

We further analyzed the proportion of obesity-induced CD153$^+$ senescent T cells via flow cytometry analysis, revealing that the proportion of splenic CD153$^+$ senescent T cells was unchanged between the ND control group and the HFD control group. In addition, neither the CD153-Alum vaccination nor the CD153-CpG vaccination affected the proportion of splenic CD153$^+$ senescent T cells (Supplementary Fig. 8A), and consistent with a previous report[20], the serum concentration of OPN was not changed by the CD153-CpG vaccination (Supplementary Fig. 8B). In contrast, the

proportion of CD153$^+$ senescent T cells in the VAT of the HFD control group was significantly increased compared with that in the VAT of the ND control group. These results suggest that obesity accelerates the increase in CD153$^+$ senescent T cells in adipose tissue under sustained HFD feeding for several months, but this increase is not systemic. Moreover, the CD153-CpG vaccination decreased the proportion of adipose CD153$^+$ senescent T cells, while the CD153-Alum vaccination did not (Fig. 4a). These results indicate that in HFD-induced obese mice, the CD153-CpG vaccination improves local adipose senescent T cells. The mRNA expression levels of proinflammatory cytokines such as IFN-γ, spp1 (as OPN) and TNF in adipose tissues also tended to be decreased in mice immunized with the CD153-CpG vaccine (Supplementary Fig. 9). To clarify whether CD153-CpG vaccine-induced antibodies have the capacity to remove CD153-expressing cells, an antibody-mediated CDC assay was performed. Lipopolysaccharide (LPS)-stimulated RAW 264.7 cells were selected[21] as the antigen for the CDC assay in the present study because of the high level at which they express CD153 (40–60%) compared with the level of CD153 expression in murine splenic or adipose cells (5–20%). The CDC assay results showed that the IgG (30, 100, 300 μg/ml) purified from mice immunized with the CD153-CpG vaccine-induced complement-dependent cell death, like the commercial anti-mouse CD153 antibody (10, 30, 100 μg/ml), while the IgG (30, 100, 300 μg/ml) purified from mice immunized with the CD153-Alum vaccine, KLH-Alum vaccine or KLH-CpG vaccine did not (Fig. 4b).

Histological analysis of VAT, kidney tissues, and lung tissues showed no significant changes in any of the groups. The VAT weight (Supplementary Fig. 10A) and average adipocyte surface area (Supplementary Fig. 10B, C) tended to be decreased in mice immunized with the CD153-CpG vaccine compared with those in HFD control group and KLH-CpG-vaccinated group mice. Obesity leads to the accumulation of macrophages in crown-like structures (CLS) around adipocytes, which is associated with chronic VAT inflammation[22,23], and CD153$^+$ senescent T cells are also localized in CLS[11]. Immunohistochemical staining for F4/80 showed that fewer F4/80$^+$ cells accumulated in the CLS of mice immunized with the CD153-CpG vaccine than in HFD control group mice (Fig. 4c). Immunohistochemical staining for CD153 also showed that the accumulation of CD153$^+$ senescent T cells in CLS was ameliorated in the CD153-CpG-vaccinated group compared with that in HFD control group mice (Fig. 4d). Of importance, TUNEL-positive cells were markedly increased, and senescent cells such asγ-H2AX-positive cells tended to be decreased in the VAT of CD153-CpG-vaccinated mice (Supplementary Fig. 11). It might be suggested that CD153-CpG vaccine induces apoptosis/cell death and ameliorates the accumulation of senescent cells. Moreover, we stained VAT, kidney tissues and

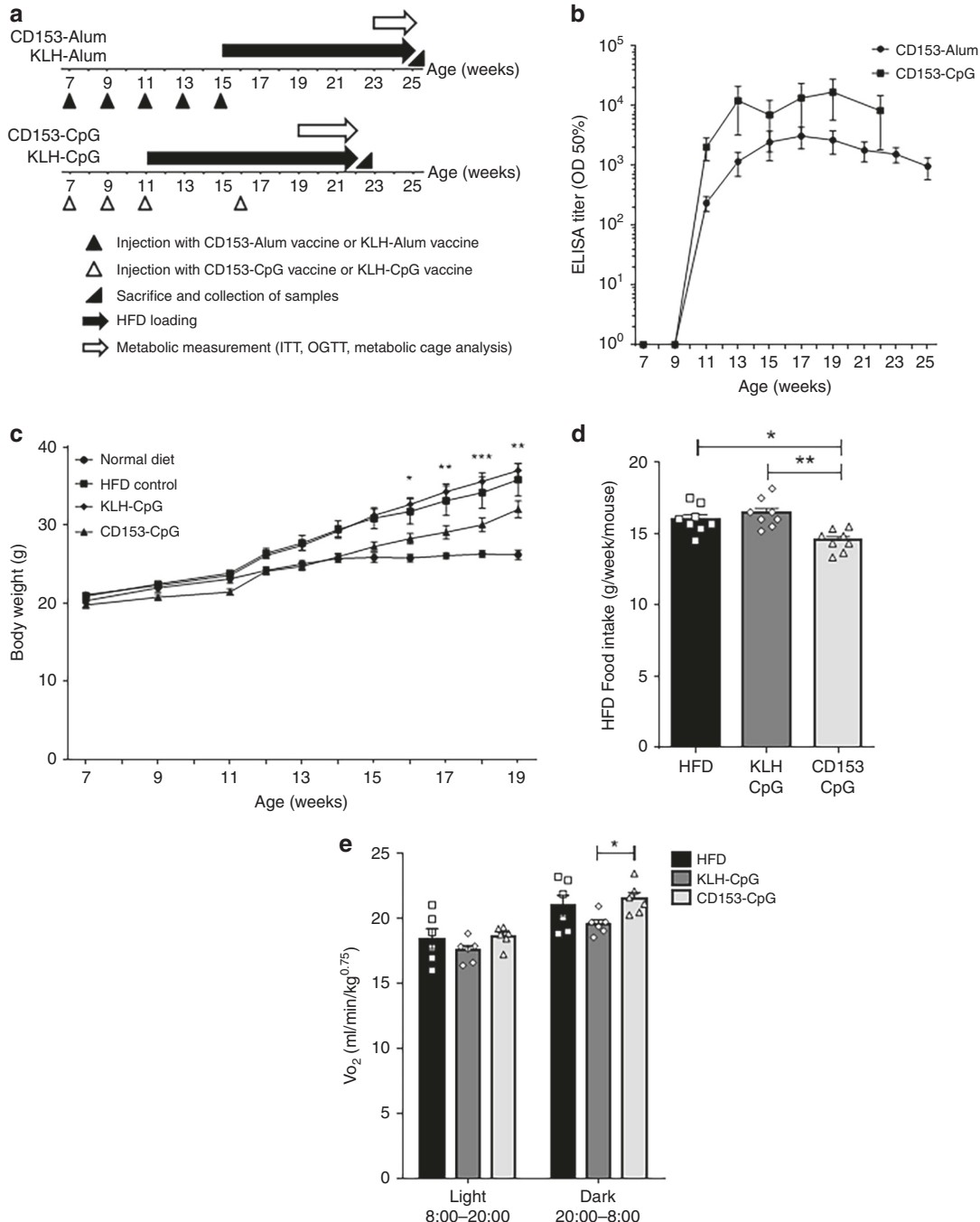

**Fig. 2 Metabolic analysis of HFD-loaded mice immunized with the CD153 vaccine. a** Time course of HFD loading and injection of vaccines. Mice immunized with the CD153-Alum vaccine or the KLH-Alum vaccine ($n = 3$ in each group) were vaccinated at the ages of 7, 9, 11, 13, and 15 weeks and fed a HFD from the age of 15 weeks. Mice immunized with the CD153-CpG vaccine or the KLH-CpG vaccine ($n = 6$ in each group) were vaccinated at the ages of 7, 9, 11, and 16 weeks and fed a HFD from the age of 11 weeks. Mice in the ND (normal diet) control group ($n = 5$) were unvaccinated and fed a ND, while mice in the HFD control group ($n = 6$) were unvaccinated and fed a HFD from the age of 11 weeks. **b** Titers against CD153-BSA in mice immunized with the CD153-Alum vaccine ($n = 3$) or the CD153-CpG vaccine ($n = 6$) during HFD loading. Titers are expressed as the half-maximal binding (OD 50%). **c** The body weights of mice in the ND control group ($n = 5$), mice in the HFD control group ($n = 6$), and mice immunized with the CD153-CpG vaccine ($n = 6$) or the KLH-CpG vaccine ($n = 6$) before metabolic measurement. Asterisk indicates the CD153-CpG group vs. the KLH-CpG group. **d** The weekly average amount of HFD intake per mouse (g/week/mouse) in mice immunized with the CD153-CpG vaccine or the KLH-CpG vaccine during HFD loading. Data were collected from 11 to 19 weeks of age ($n = 8$ in each group). **e** The average VO₂ values during the light (8:00–20:00) and dark (20:00–8:00) periods in mice in the HFD control group ($n = 6$) and in mice immunized with the CD153-CpG vaccine ($n = 6$) or the KLH-CpG vaccine ($n = 6$) at the age of 21–22 weeks. All the data are expressed as the mean ± SEM. Statistical evaluation was performed by two sided; Tukey's multiple comparison test (**c**, **d**); Bonferroni correction, HFD vs. KLH-CpG vs. CD153-CpG (**e**); *$p < 0.05$; **$p < 0.01$; ***$p < 0.001$.

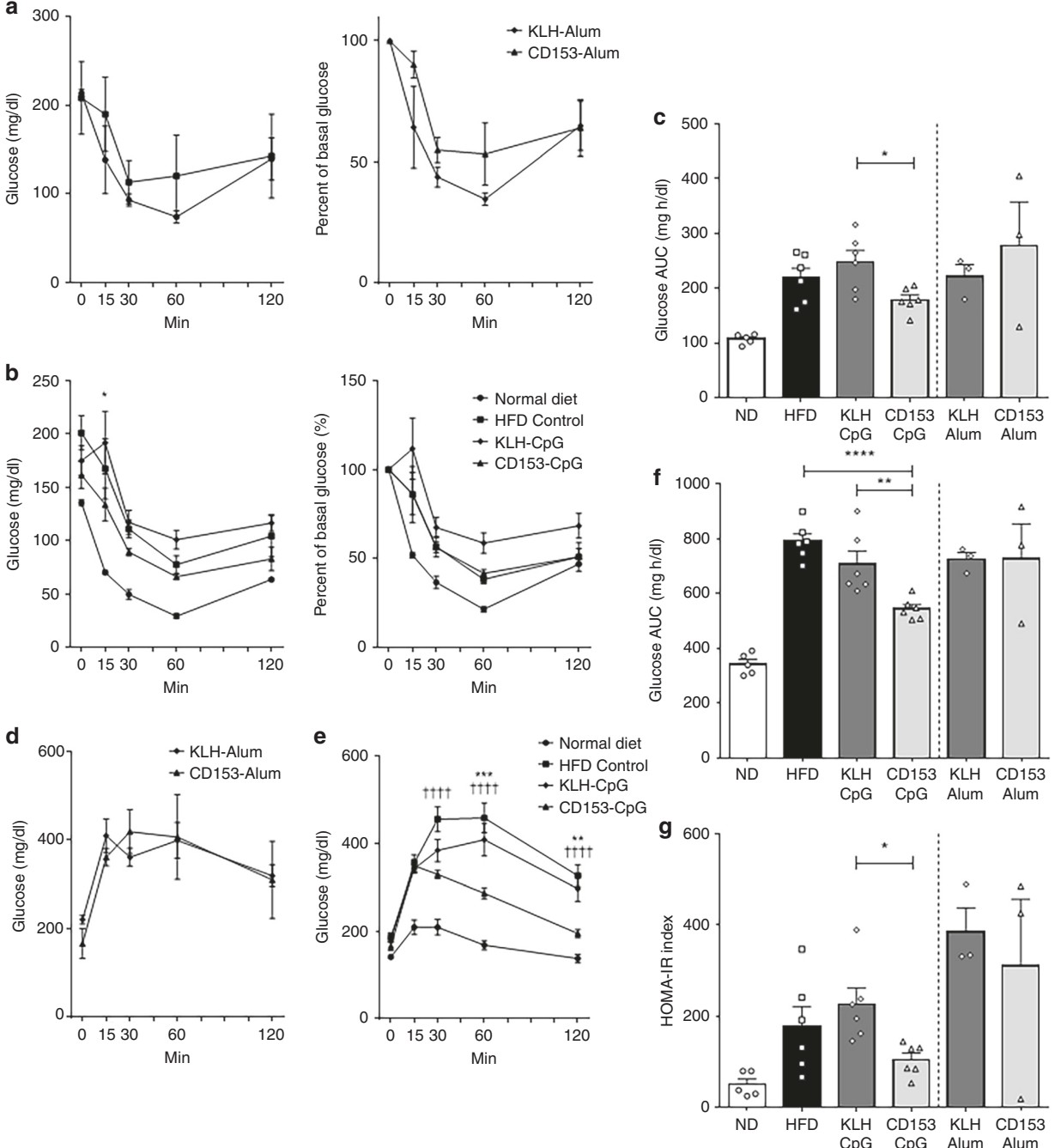

**Fig. 3 Glucose tolerance and insulin sensitivity in HFD-loaded mice immunized with CD153 vaccine. a–c** Blood glucose concentrations and percent basal glucose in mice immunized with the Alum-conjugated vaccine (CD153-Alum or KLH-Alum) (**a**; $n = 3$ in each group) or the CpG-conjugated vaccine (CD153-CpG or KLH-CpG) (**b**; $n = 6$ in each group) as determined by the intraperitoneal insulin tolerance test (ipITT). Mice in both the ND (normal diet, $n = 5$) and HFD (high-fat diet, $n = 6$) control groups were treated in a manner similar to the vaccinated group. Asterisk indicates the CD153-CpG group vs. the KLH-CpG group. **c** Area under the curve (AUC) of blood glucose levels as determined by the ITT. The AUC was estimated using the trapezoidal rule. **d–f** Blood glucose concentrations in mice immunized with the Alum-conjugated vaccine (**d**; $n = 3$ in each group) or the CpG-conjugated vaccine (**e**; $n = 6$ in each group) as determined by the oral glucose tolerance test (OGTT). Mice in both the ND (normal diet, $n = 5$) and HFD (high-fat diet, $n = 6$) control groups were treated in a manner similar to the vaccinated group. Asterisk indicates the CD153-CpG group vs. the KLH-CpG group. Dagger indicates the CD153-CpG group vs. the HFD control group. **f** AUC of blood glucose levels as determined by the OGTT. The AUC was estimated using the trapezoidal rule. **g** HOMA-IR index after 6 h of fasting. Alum-conjugated vaccine group, $n = 3$ in each group; CpG-conjugated vaccine group, $n = 6$ in each group; ND control group, $n = 5$; HFD control group, $n = 6$. All the data are expressed as the mean ± SEM. Statistical evaluation was performed by two sided; unpaired two-tailed t-test, KLH-Alum vs. CD153-Alum (**c**, **f**, **g**); Bonferroni correction, HFD vs. KLH-CpG vs. CD153-CpG (C, F and G); Tukey's multiple comparison test (**a**, **b**, **d**, **e**); *$p < 0.05$; **$p < 0.01$; ***$p < 0.001$; ****$p < 0.0001$; ††††$p < 0.0001$.

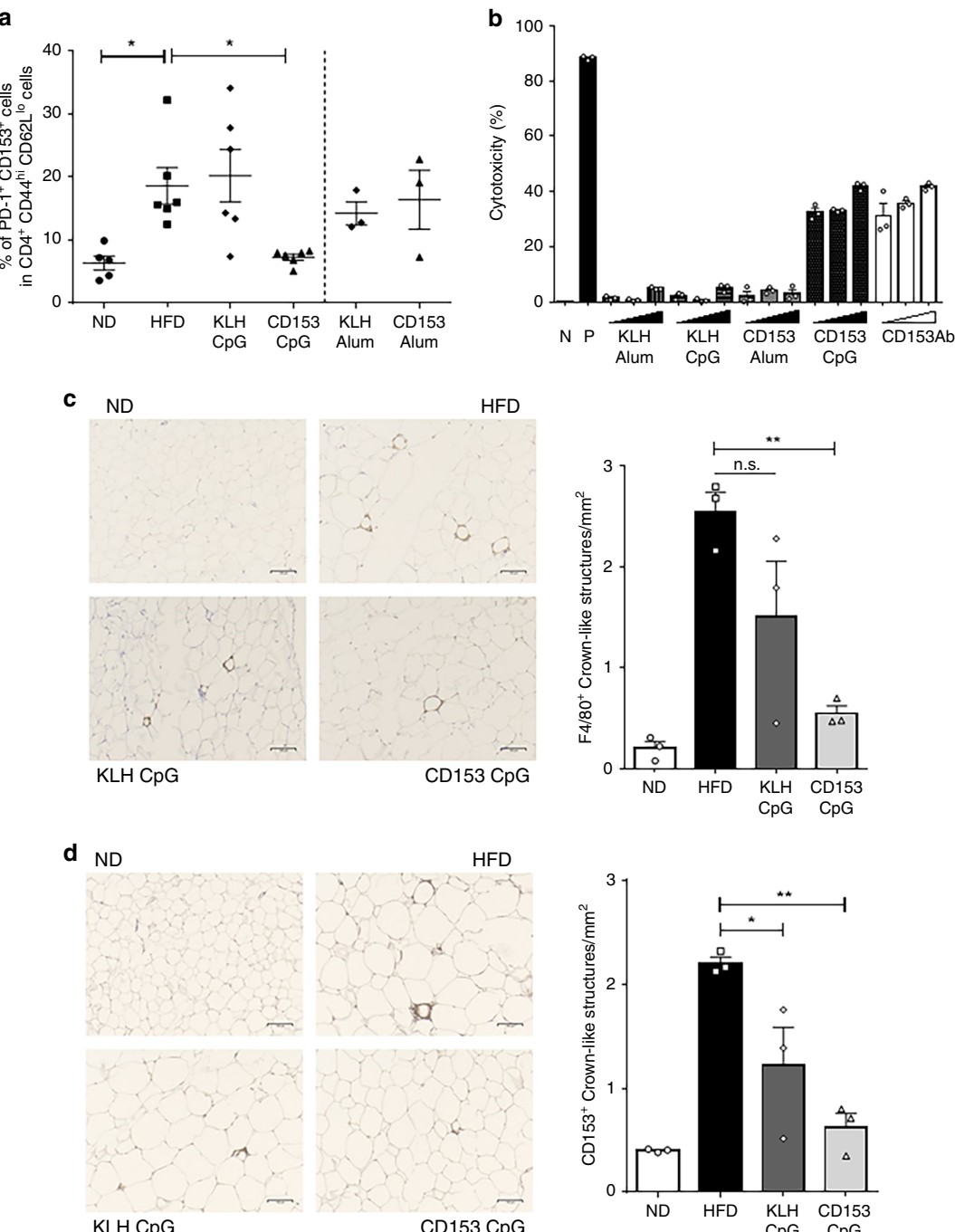

**Fig. 4 Effect of the CD153-CpG vaccine on CD153+ senescent T cells induced by HFD loading. a** The proportions of senescent T cells induced by HFD loading in the VAT of mice immunized with the Alum-conjugated vaccine or the CpG-conjugated vaccine. Senescent T cells were defined as PD-1+ CD153+ cells in CD4+, CD44hi, CD62Llo cells. Alum-conjugated vaccine group, $n = 3$ in each group; CpG-conjugated vaccine group, $n = 6$ in each group; ND control group, $n = 5$; HFD control group, $n = 6$. **b** The CDC activity of the CD153 vaccine-induced antibody was assessed ($n = 3$ each sample) using $2 \times 10^4$ LPS-stimulated RAW 264.7 cells. The vaccine-induced IgG antibodies used were purified from the sera of HFD-loaded mice at the age of 22–25 weeks immunized with the CD153-Alum vaccine, CD153-CpG vaccine, KLH-Alum vaccine or KLH-CpG vaccine. N, no stimulation with antibody; P, 100 μg/ml anti-mouse MHC class I antibody (H-2Kd/H-2Dd); closed right triangle, the concentrations of purified IgG antibodies; opened right triangle, the concentrations of anti-mouse CD153 antibody (10, 30, 100 μg/ml, respectively). **c, d** VAT were collected from the four groups of mice (ND control group, HFD control group, KLH-CpG vaccine group, and CD153-CpG vaccine group; $n = 3$ in each group) according to the time course of HFD loading and injection of vaccines. **c** Quantification of macrophages in the crown-like structures among the four groups. HFD vs. KLH-CpG, $p = 0.11$; HFD vs. CD153-CpG, $p = 0.0041$; KLH-CpG vs. CD153-CpG, $p = 0.15$. 40–50 fields per sample, F4/80 (brown); scale bars: 100 μm. **d** Quantification of CD153-positive cells in the crown-like structures among the four groups. HFD vs. KLH-CpG, $p = 0.024$; HFD vs. CD153-CpG, $p = 0.0015$; KLH-CpG vs. CD153-CpG, $p = 0.20$. 40–50 fields per sample, CD153 (brown); scale bars: 100 μm. All the data are expressed as the mean ± SEM. Statistical evaluation was performed by two-sided; unpaired two-tailed t-test, KLH-Alum vs. CD153-Alum (**a**); Bonferroni correction, ND vs. HFD (**a**), HFD vs. KLH-CpG vs. CD153-CpG (**a, c, d**); *$p < 0.05$; **$p < 0.01$; n.s. not significant.

lung tissues with an anti-mouse IgG antibody to evaluate whether antibody immunocomplexes were present (Supplementary Figs. 12 and 13). In VAT, there was no positive staining for IgG except for a slight positive stain in the CLS of mice immunized with the CD153-CpG vaccine, including ND-CD153-CpG vaccine group. In lung tissues, alveolar wall deposition of the IgG antibody was observed in mice immunized with the KLH-CpG and CD153-CpG vaccines, but no apparent differences in the thickness of the alveolar wall or infiltration of inflammatory cells were observed. Similarly, in kidney tissues, glomerulus deposition of the IgG antibody was observed in mice immunized with the KLH-CpG and CD153-CpG vaccines, but no apparent differences in the thickness or delicacy of glomerular capillary loops were observed.

## Discussion

Recently therapeutic self-antigen vaccines have been developed, such as angiotensin II for hypertension[24] and DPP4 for diabetes[25], in several models that are unlike the standard vaccines for infectious diseases or cancer. The basic concept of our therapeutic vaccines is to induce antibody production, whereas standard vaccines are aimed at activating both cytotoxic T cells and antibody production. In our immune tolerance system, the immune reaction against self-antigens is tightly limited by blocking the activation of self-reactive T cells, but self-reactive B cells remain active and can be provoked by efficient T-cell activation. Peripheral T-cell tolerance usually inactivates T cells by the induction of 'anergy'; however, this tolerance could be disrupted by cotreatment with adjuvants, leading to activation of the innate immune system. Therefore, adjuvants also play important roles in activating self-reactive B cells to induce antibody production. In terms of adjuvant selection, standard vaccines use Th1-directed adjuvants (i.e., CpG oligonucleotide) to induce mouse IgG2 (human IgG1) antibody production[14,15], while our previous therapeutic vaccines require Th2-directed adjuvants (i.e., Alhydrogel) to induce mouse IgG1 (human IgG2) antibody production. Because the mouse IgG1 (human IgG2) antibody does not have effector function, which includes antibody-dependent cell-mediated cytotoxicity (ADCC) and CDC, it can be a blocking antibody suitable for neutralizing molecular functions without exerting toxicity. However, our current idea is to utilize the modified vaccine, which selectively induces an antibody with effector functions, to prevent the accumulation of senescent cells by inducing apoptosis/cell death. Therefore, the CpG adjuvant, but not the Alum adjuvant, is suitable for pushing cells in the Th1 direction and inducing the mouse IgG2 antibody. Indeed, the mouse IgG2 antibody accompanied by CDC was mainly produced in CD153-CpG-vaccinated mice and successfully reduced the number of senescent T cells.

We also designed a human CD153#D peptide vaccine selected from human CD153 peptide (71–80 aa) homologous to mouse CD153#D peptide (76–85 aa) (Supplementary Fig. 14A) to examine the cross-reactivity with human and mouse. The human CD153#D-Alum vaccine was administered to 7-week-old male C57BL/6J mice three times at 2-week intervals as well as mouse CD153#D-Alum vaccine, and human or mouse CD153 antibody production was evaluated by ELISA. We found the antibody induced by the human CD153#D vaccine strongly reacted with human CD153#D-BSA and recombinant human CD153 (rhCD153), while it did not react with mouse CD153#D-BSA and rmCD153 (Supplementary Fig. 14B, C). The antibody induced by the mouse CD153#D vaccine hardly reacted with rhCD153 (Supplementary Fig. 14D), unlike with rmCD153.

In the present study, we successfully developed a modified vaccine to prevent the accumulation of senescent T cells expressing the surface marker CD153 and ameliorate obesity-related metabolic disorders. However, our study has a few limitations for designing other senotherapy applications. Because the antibody cannot recognize intracellular molecules, the ability of the target protein in our vaccine to delete senescent cells is limited to whether the cells express a specific surface marker. In addition, removal by CDC or ADCC may induce inflammation in each tissue[26], and the target protein should be expressed at a lower level to minimize adverse effects. Therefore, the selection of target proteins is critically important for applying this modified vaccine to senotherapy. Other study limitations include the evaluation method of cellular senescence; cellular senescence of SA-T cells was assessed only by higher expression of SA–β-gal and γ-H2AX. Further studies are warranted to determine SA-T cells' potency as senescent cells and their contribution to several diseases.

CD153$^+$ T cells are reported as a mediator of CD4 T-cell-dependent control of *Mycobacterium tuberculosis* infection in the lung tissue[27]. Although tumor necrosis factor alpha (TNF-α) inhibitors is similarly associated with an increased risk of tuberculosis infection, screening, and treatment for latent tuberculosis infection in patients is effective to reduce the incidence of tuberculosis[28]. Toward clinical application of CD153-CpG vaccine, the safety evaluation and management should be further discussed based on these previous evidence. Here, we propose that the CD153-CpG vaccine might be an optional tool for senolytic therapy, and further safety evaluation and management will be required toward clinical application.

## Methods

**Vaccine design and peptide synthesis**. Based on high antigenicity analysis of the three-dimensional predicted structure and epitope information, five different antigenic peptides were selected from the amino acid sequence of mouse CD153 (Supplementary Fig. 1A). The N-terminus of the peptide was conjugated to KLH (Enzo Life Sciences Inc., Farmingdale, NY, USA) as a carrier protein, and the synthetic peptide was purified by reverse-phase HPLC (>98% purity) (Peptide Institute Inc., Osaka, Japan.) The CD153 peptide vaccine was reconstituted at 0.5–1 mg/ml of the CD153 peptide and at 5–10 mg/ml of the KLH in sterile PBS.

**Animals**. All animal experimental procedures were reviewed and approved by the Institutional Animal Committee at the Department of Veterinary Science of Osaka University School of Medicine and performed in accordance with guidelines for animal experimentation at research institutes (Ministry of Education, Culture, Sports, Science and Technology, Japan), guidelines for animal experimentation at institutes (Ministry of Health, Labor and Welfare, Japan), and guidelines for the proper conduct of animal experiments (Science Council of Japan). Seven or eight-week-old male C57BL/6J mice and 8-week-old female C57BL/6N mice were purchased from CLEA Japan Inc. and housed in a temperature-, humidity- and light cycle-controlled facility (23 ± 1 °C; 55 ± 10%; light, 8:00–20:00; dark, 20:00–8:00). Mice had free access to food and water except for mice under pair-feeding condition. C57BL/6J mice were fed either a ND (MF, 12.8 kcal% fat; Oriental Yeast Co., Ltd) or a HFD (D12492, 60 kcal% fat; Research Diets Inc.), and C57BL/6N mice were fed a ND.

**Vaccination schedule**. A single dose of the CD153 vaccine was prepared as a mixture of CD153-KLH peptide solution (30 µg of the CD153 peptide and 200–300 µg of KLH) and adjuvant solution. A single dose of the KLH vaccine was prepared as a mixture of KLH (200–300 µg) and adjuvant solution. The adjuvant solution contained 30 µl of Alhydrogel (CD153-Alum, KLH-Alum; InvivoGen) or 10 µg of CpG ODN 1585 (CD153-CpG, KLH-CpG; Invivogen). In the TLR7 ligand administration study, male C57BL/6J mice and female C57BL/6N mice were vaccinated subcutaneously with the CD153-CpG vaccine or the KLH-CpG vaccine at the age of 8, 10, and 12 weeks. In the HFD loading study, male C57BL/6J mice were vaccinated subcutaneously with the CD153-Alum vaccine or the KLH-Alum vaccine at the ages of 7, 9, 11, 13, and 15 weeks or with the CD153-CpG or KLH-CpG vaccine at the ages of 7, 9, 11, and 16 weeks. Serum was collected from the tail vein, and the anti-CD153 antibody titer was determined by ELISA.

**TLR7 ligand administration**. Twelve-week-old male C57BL/6J mice and female C57BL/6N mice were intraperitoneally injected with R848 (TLR7 ligand; InvivoGen) three times per week. The mice sacrificed at the age of 16 weeks were injected with 5 µg of R848 for 4 weeks. The mice sacrificed at the age of 18 weeks were injected with 5 µg of R848 for 4 weeks and injected with 10 µg of R848 for an additional 2 weeks.

**Cell lines and culture conditions**. The murine macrophage cell line RAW 264.7 was obtained from the American Type Culture Collection (ATCC), grown in Dulbecco's Modified Eagle's Medium (DMEM; Nacalai Tesque, Kyoto, Japan) supplemented with 10% (v/v) heat-inactivated FBS and stimulated for 24 h with 2 μg/ml LPS (L4391, from *Escherichia coli* O111:B4; Sigma Aldrich, MO, USA) prior to harvest. The cell lines were incubated at 37 °C and 5% CO₂ according to ATCC animal cell culture guidelines.

**Enzyme-linked immunosorbent assay**. The serum anti-CD153 antibody titer was quantified by ELISA. CD153-BSA conjugate (Peptide Institute Inc., Osaka, Japan) or recombinant mouse CD153 protein (carrier-free; R&D Systems, MN, USA) was coated at a 10 μg/ml concentration and diluted in 50 mM carbonate buffer overnight at 4 °C on 96-well ELISA plates (MaxiSorp Nunc, Thermo Fisher Scientific K.K., Japan). After blocking with PBS containing 5% skim milk, the sera were serially diluted from 100- to 325,000-fold in blocking buffer, added to each well, and incubated overnight at 4 °C. After washing each well with 0.05% PBS Tween-20 (PBS-T), the cells were incubated with horseradish peroxidase (HRP)-conjugated antibodies specific for mouse IgG (1:1000; GE Healthcare, UK) for 3 h at room temperature. For the IgG subclass determination assay, anti-mouse IgG subclass-specific HRP-conjugated antibodies (1:1000; IgG1, IgG2b, IgG2c and IgG3, Abcam) were used. After washing the wells with PBS-T, color was developed with the peroxidase chromogenic substrate 3,3′,5,5′-tetramethylbenzidine (TMB; Sigma Aldrich, MO, USA), and the reactions were terminated with 0.5 N sulfuric acid. The absorbance was measured at 450 nm using an iMark microplate absorbance reader (Bio-Rad, CA, USA) and analyzed with MPM6 version 6.1 (Bio-Rad). The half-maximal antibody titer was analyzed with ImageJ version 1.48 and determined according to the highest value in the dilution range of each sample. The serum concentration of OPN was also determined using the Mouse Osteopontin DuoSet ELISA kit (R&D Systems) according to the manufacturers' instructions.

**Western blot analysis**. Briefly, 30 or 100 ng of rmCD153 (rmCD153; R&D Systems) and 100 ng of rmOPN (used as a negative control; R&D Systems) were electrophoretically separated by SDS/PAGE and blotted onto hydrated Immobilon-P PVDF transfer membranes (Merck Millipore Ltd.). Precision Plus Protein WesternC Standards (Bio-Rad) was applied to each membrane as a marker. Sera from mice immunized with the CD153#D-Alum vaccine or KLH vaccine were diluted 500-fold. The commercial anti-CD153 antibody was diluted to 0.05 μg/ml. After incubation with HRP-conjugated IgG antibodies (GE Healthcare) diluted 2000-fold for 1 h, a chemiluminescent signal visualized by Chemi-Lumi One L (Nacalai Tesque) was detected with a LAS 1000 instrument (Fuji Film) and analyzed with Multi Gauge software version 3.2.

**CDC assay**. Total IgG antibodies were purified from sera pooled from immunized mice using a 50% saturated ammonium sulfate solution, Zeba Spin Desalting Columns and the Melon Gel IgG Spin Purification Kit (Thermo Fisher Scientific, IL, USA). For the CDC assay, $2 \times 10^4$ LPS-stimulated RAW 264.7 cells suspended in DMEM supplemented with 1% FBS were mixed with purified IgG antibodies at various concentrations (30, 100, 300 μg/ml) or positive control antibodies and incubated at 4 °C for 1 h. The positive control antibodies included an anti-mouse CD153 antibody (functional grade, RM153; eBioscience) and an anti-mouse major histocompatibility complex (MHC) class I antibody (H-2Kd/H-2Dd, functional grade, 34-1-2S; eBioscience). Low-Tox-M rabbit complement (Cedarlane, Hornby, Canada) was then added to a final concentration of 10% and incubated at 37 °C for 1 h. After centrifugation at $300 \times g$ at 4 °C for 5 min, 1 μl of the 7-AAD viability staining solution (BD Biosciences) was added to the resuspended solutions. Cell death was evaluated as the percentage of 7-AAD-positive cells using flow cytometry analysis.

**Metabolic measurements**. Insulin sensitivity was assessed by the ipITT after 4 h of fasting. Blood glucose concentrations were measured before and 15, 30, 60, and 120 min after an intraperitoneal injection of 0.75 U/kg recombinant human insulin (Humulin R; Eli Lilly Japan K.K., Japan). Glucose tolerance was assessed by the OGTT after 6 h of fasting. Blood glucose concentrations were measured before and 15, 30, 60 and 120 min after oral administration of 2.0 g/kg glucose[29]. HOMA-IR was calculated as an index for insulin resistance as follows: fasting serum glucose (mmol/L) × fasting serum insulin (pmol/L)/22.5. The serum concentration of insulin was determined using the Mouse Insulin ELISA kit (Mercodia) according to the manufacturers' instructions. Oxygen consumption was measured with an O₂/CO₂ metabolism measuring system for small animals (MK-5000RQ; Muromachi Kikai, Tokyo, Japan), and the results were analyzed using MMS-2 software (Muromachi Kikai). Each mouse was placed in an airtight chamber maintained at 25 °C with 0.50 L/min of air flow for more than 24 h, and the oxygen consumption was normalized by kilogram$^{0.75}$ body weight.

**Flow cytometry analysis**. Spleen samples from immunized mice were triturated and suspended in staining buffer (BD Biosciences, San Diego, CA, USA). VAT were collected from epididimal fat pads, minced into fine pieces and incubated in a digestion buffer composed of Hank's Balanced Salt Solution (HBSS) in 10% FBS, 100 μg/ml DNase I and 200 U/ml collagenase type I (Worthington, Lakewood, NJ, USA) at 37 °C for 1 h while shaking. Digested tissue was centrifuged at $1000 \times g$ for 10 min at 4 °C and resuspended in staining buffer. Red blood cells were removed from the suspensions of splenocytes and adipose stromal vascular fractions (SVF) using ACK erythrocyte-lysing buffer (Gibco, Grand Island, NY, USA). After lysing erythrocytes, the suspended splenocytes and SVF were filtered through a 70-μm filter, centrifuged at $1000 \times g$ for 10 min at 4 °C and resuspended in staining buffer. After blocking Fc-receptors with an anti-mouse CD16/32 antibody (mouse Fc-receptor blocker; BD Biosciences) for 20 min at 4 °C, cells were stained with a mixture of fluorescently labeled antibodies at 4 °C for 40 min in the dark. The antibodies used were specific to CD4-FITC (RM4-4), CD44-PE (IM7), CD153-BV421 (RM153) (BD Biosciences), CD62L-APC-Cy7 (MEL-14), and PD-1-APC (29F.1A12) (Bio Legend). A 7-AAD viability staining solution was added to exclude dead cells. Flow cytometry analysis was performed using BD FACS Canto II (BD Biosciences), and the results were analyzed using BD FACS Diva software version 8.0.1 (BD Biosciences). In addition, intracellular staining of phosphorylated histone H2AX at serine 139 (defined as γ-H2AX) was also performed using a fixation/permeabilization solution kit (BD Biosciences) according to the manufacturer's recommendations. Blocking Fc-receptors and staining cells with a mixture of fluorescently labeled antibodies, CD4-APC-Cy7 (RM4-5, BD Biosciences), CD44-PE, CD153-BV421 and PD-1-APC, was performed before intracellular staining of γ-H2AX-Alexa Fluor 488 (2F3, Bio Legend).

**Cellular senescence assay**. SA-β-gal assay was performed using a cellular senescence detection kit (SPiDER-βGal, Dojindo). Blocking Fc-receptors and staining cells with a mixture of fluorescently labeled antibodies, CD4-APC-Cy7, CD44-PE, CD153-BV421, and PD-1-APC, was performed after SA-β-gal assay.

**RNA extraction and real-time quantitative PCR**. For quantitative PCR, total RNA was extracted from collected VAT samples using a TRIzol plus RNA purification kit (Thermo Fisher Scientific, USA), in which on-column DNase digestion was included. The cDNA was then synthesized using a high-capacity cDNA reverse transcription kit (Applied Biosystems, MA, USA) on T100 Thermal Cycler (Bio-Rad Inc.). Quantitative real-time PCR was performed using a 7900HT fast real-time PCR system (Applied Biosystems) and a real-time PCR master mix (TOYOBO) with gene-specific primers in 15 μl of mixture following the manufacturers' instructions. The following hydrolysis probes (TaqMan® MGB Probe, Applied Biosystems) were used for expression analysis: *Gapdh*, Mm99999915_g1; *Ifn-γ*, Mm01168134_m1; *Il-1β*, Mm99999061_mH; *Il-6*, Mm00446190_m1; *Spp1*, Mm00436767_m1; *Tnf*, Mm99999068_m1. Quantitative data were normalized using *Gapdh* as an endogenous reference gene and calculated using the relative standard curve method. These data were analyzed using 7900HT SDS version 2.4.1 (Applied Biosystems).

**Immunohistochemistry**. For paraffin sections, murine VAT, kidney and lung samples were fixed with 10% neutral buffered formalin and paraffin-embedded. Hematoxylin and eosin staining was performed on VAT, kidney tissues, and lung tissues. After deparaffinization and rehydration, immunohistochemical staining for F4/80 (CI:A3-1, Abcam), CD153 (Biorbyt) or γ-H2AX (Abcam) was performed on VAT, and immunohistochemical staining for IgG (only a secondary antibody was used) was performed on VAT, kidney tissues and lung tissues. For antigen retrieval, incubation with Proteinase K (Agilent Technologies) was performed for 5 min for F4/80 and IgG staining, and incubation with HistoVT One (Nacalai Tesque) was performed for 30 min at 90 °C for CD153 and γ-H2AX staining. For enzymatic detection, an avidin-biotin complex kit (VECTASTAIN Elite ABC kit, Vector Laboratories) was used for CD153 and γ-H2AX, and MAX-PO(R) (Nichirei Bioscience), a polymer based detection kit, was used for F4/80 and MAX-PO (M) (Nichirei Bioscience) was used for IgG staining according to the manufacturer's recommendations. Endogenous peroxidase was blocked with 0.3–0.6% H₂O₂ in methanol before enzymatic detection. In addition, TUNEL staining was performed on VAT. After incubation with Proteinase K for 5 min, ApopTag Peroxidase In Situ Apoptosis Detection Kit (Millipore) was used according to the manufacturer's recommendations. All bright-field micrographs were acquired on a FSX100 system (Olympus) and analyzed with FSX-BSW version 3.2.0.0 (Olympus) and ImageJ version 1.48.

**Statistical analyses**. All values are presented as the mean ± SEM. The statistical significance of differences between two groups was assessed by two-tailed unpaired *t*-test. Differences among multiple groups were assessed by analysis of variance test, Tukey's multiple comparison test or Bonferroni correction. A difference was considered statistically significant when $p < 0.05$. Statistical analysis was performed using Prism GraphPad version 6.07 (GraphPad Software).

**Reporting summary**. Further information on research design is available in the Nature Research Reporting Summary linked to this article.

## Data availability

The data that support this study are available from the corresponding authors upon reasonable request. Full scans of the gels and blots are available in Supplementary Fig. 15.

The sequences of the primers and probes used for real-time PCR measurement are available in Supplementary Table 1. The source data underlying Figs. 1a–e, 2b–e, 3a–g and 4a–d and Supplementary Fig. 1B, 3, 4B, 5A, B, 6A–C, 7A–E, 8A, B, 9, 10A, B, 11 and 14B–D are provided as a Source Data file.

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

## Acknowledgements

This work was partially supported by JSPS KAKENHI Grant Number JP15K15310. We thank all members of the Department of Health Development and Medicine for supporting this project.

## Author contributions

S.Y. and H.N. wrote the manuscript. S.Y., H.N., H.T., R.M. and H.R. designed the study with discussion. S.Y., H.H., Y.I., J.S., A.T., T.K, and M.S. performed the experiments and analyzed the data.

## Competing interests

Department of Health Development and Medicine is endowed department supported by Anges, Daicel, and Funpep. Department of Clinical Gene Therapy is financially supported by Novartis, AnGes, Shionogi, Boeringher, Fancl, Saisei Mirai Clinics, Rohto and Funpep. R.M. is a stockholder of FunPep and AnGes. H.T. and A.T. are employees and stockholders of FunPep. The funder (Funpep) provided support in the form of salaries for authors (H.T. and A.T.) but did not have any additional role in the study design, data collection and analysis, decision to publish, or preparation of the manuscript. All other authors declare no competing interests.
