## [Peer Review File · Nature Communications]

Reviewers' Comments:

Reviewer #1:

Remarks to the Author:

In this manuscript, the authors present what I believe is the first evidence illustrating the development of a vaccine against senescent cells. Specifically, they target senescent T cells that accumulate in visceral adipose tissue during obesity. These senescent cells are defined by the surface markers CD4⁺ CD44^{high} CD62L^{low} PD-1⁺ CD153⁺. Vaccination was with a CD153 peptide conjugated to KLH and resulted in increased anti-CD153 antibody production in mice. The authors found the antibody lowered senescent T cells in adipose tissue of mice fed a high fat diet for 10 weeks. This was accompanied by improved glucose tolerance and insulin sensitivity. They provide in vitro evidence (complement-dependent cytotoxicity assay) to support the hypothesis that the CD153 antibodies evoked have the capacity to remove CD153⁺ cells. In vivo, they show fewer crown-like structures around adipocytes, a marker of white adipose tissue inflammation.

The experiments are carefully executed, and numerous controls are used, including using KLH alone to immunize mice, using different adjuvants, and comparing endogenous anti-CD153 to a commercially available antibody.

Overall, this is an intriguing proof-of-concept study. Applications are limited in that CD153⁺ cells do not increase significantly in visceral fat with normal aging (Fig. 1D) and CD153⁺ T cells are important for preventing lung infection (Nat Microbio 2018). Thus, the utility of a CD153-CpG vaccine as a senolytic therapy has both safety and utility issues. The last sentence of the discussion might be tempered accordingly.

In addition, in Figure 4, panels C & D are expressed as percentage of crown-like structures around adipocytes and CD153⁺CLS. The data should be expressed as actual values. N = 3 mice per group. How many sections? How many fields counted? How many actual CLS observed? And how many stained positive for CD153? This is important to provide an indication as to what extent senescent T cells must be cleared to have a physiological impact.

Reviewer #2:

Remarks to the Author:

In the manuscript entitled "The CD153 vaccine with CpG adjuvant is a novel serotherapeutic option for removing senescence-associated T cells from visceral adipose tissues" authors describe development of the vaccine to target CD153 expressing, senescent associated (SA)- T lymphocytes. Infiltration of immune cells in fat is a hallmark of obesity related insulin resistance. It has been demonstrated previously that among other immune cell types CD153⁺PD-1⁺CD44^{hi}CD4⁺ T cells accumulate in the adipose tissue with obesity and contribute to obesity induced insulin resistance. Attempt to clear or prevent accumulation of SA-T lymphocytes in fat could be an attractive approach to inhibit inflammation and improve metabolic dysfunction associated with obesity.

However, the manuscript has major issues which need to be addressed before it is considered for publication.

Mice vaccinated with CD-153-CpG vaccine described in the paper have decrease food consumption and body weight compared to mock vaccinated and high fat feed controls. Decrease food consumption in mice indicates stress or illness, unless intervention impacts CNS. As authors stated CD-153-CpG vaccinated mice after 11 weeks of high fed feeding have improved insulin glucose tolerance and insulin sensitivity and less CD153⁺SA T lymphocytes in the visceral fat. Many study endpoints described in the experiment with high fat feeding are sensitive to food intake, body weight and the level of adiposity, and it is important to clarify whether reported outcomes involving these measures are truly reflective of efficacy of vaccine or are a result of an adverse event/ toxicity of the immunization.

To clarify these authors should perform "pair feeding", and have additional control groups in the experiment. They need to vaccinate chow fed mice and look if there is any adverse or off target effect.

Vaccination against CD153 positive lymphocytes might be risky and has very limited if any path to translation. CD153 expressing CD4+ lymphocytes play an important role in modulating immune response and fight infection. Sallin MA, Kauffman KD, Riou C, Du Bruyn E, Foreman TW, Sakai S, Hoft SG, Myers TG, Gardina PJ, Sher A, Moore R, Wilder-Kofie T, Moore IN, Sette A, Lindestam Arlehamn CS, Wilkinson RJ, Barber DL. Host resistance to pulmonary Mycobacterium tuberculosis infection requires CD153 expression. *Nature Microbiology* 2018/11/01;3(11):1198-205. Vaccine which has prolonged sustainable effect against this ligand might have serious adverse effects on health. Author's should quote this paper and address this issue in the limitation of the approach.

There are publications showing beneficial effects of senescent cell clearance on metabolic health. Authors should quote these publications to make the paper less superficial

Xu M, Palmer AK, Ding H, Weivoda MM, Pirtskhalava T, White TA, Sepe A, Johnson KO, Stout MB, Giorgadze N, Jensen MD, LeBrasseur NK, Tchkonina T, Kirkland JL. Targeting senescent cells enhances adipogenesis and metabolic function in old age. *Elife* 2015 Dec 19;4:e12997.

Schafer MJ, White TA, Evans G, Tonne JM, Verzosa GC, Stout MB, Mazula DL, Palmer AK, Baker DJ, Jensen MD, Torbenson MS, Miller JD, Ikeda Y, Tchkonina T, van Deursen JM, Kirkland JL, LeBrasseur NK. Exercise Prevents Diet-Induced Cellular Senescence in Adipose Tissue. *Diabetes* 2016 Jun;65(6):1606-15.

Reviewer #3:

Remarks to the Author:

In the manuscript "The CD153 vaccine with CpG adjuvant is a novel serotherapeutic option for removing senescence-associated T cells from visceral adipose tissues", the authors show that vaccination can delete cells that contribute to a senescent phenotype and improve obesity-induced metabolic disorders. I was asked to comment on the immunological aspects of this manuscript. The vaccination protocol and assessment of the response to vaccination are all appropriate and standard for the field. Quantification of antibody production and function (in a cytotoxic assay) are all done properly as is the flow cytometric analysis of cell populations. The only issue that I have is with the animals used in these studies. In the TLR7 ligand administration study, female mice were used, while in the HFD loading study, male mice were used. It is well known that males and females have different immune responses to vaccination. I would have liked to see both sexes be used for both experiments. Aside from this point, this is a well done, nicely presented study.

Reviewer #4:

Remarks to the Author:

Summary: The authors developed a vaccine with a peptide from CD153 and CpG 1585. The vaccine boosted anti-CD153 antibodies, in particular several IgG subclasses, for a prolonged period. The authors then demonstrated that the CD153-CpG vaccine was able to reduce senescence-associated T cells (SA-T cells), which express CD153, in visceral adipose tissue of obese mice, most likely mediated by IgG2 antibodies against CD153.

Overall Critique: This work is important and the data are convincing that the vaccine has potential as a therapy to deplete SA-T cells. The approach to develop the vaccine is not particularly novel, although the application is novel. I only have a few concerns.

Concerns:

1. Why did the authors chose CpG 1585 (Invivogen), a class A CpG, rather than a Class B or C CpG? It would be interesting to compare each of these classes of TLR9 agonists in your model.

2. Is the mouse CD153#D peptide homologous in humans? In other words, could this approach be easily translated to vaccine development for humans?
3. Can you provide statistical analyses for Figure 1A and Figure 1C?
4. Why only 3 mice in Figure 1F?

Response to Reviewer #1 (MD ID: NCOMMS-19-00198-T)

Thank you very much for your very important comments. According to your suggestions, we have certainly tried our best to revise the manuscript.

1. Overall, this is an intriguing proof-of-concept study. Applications are limited in that CD153+ cells do not increase significantly in visceral fat with normal aging (Fig. 1D) and CD153+ T cells are important for preventing lung infection (Nat Microbio 2018). Thus, the utility of a CD153-CpG vaccine as a senolytic therapy has both safety and utility issues. The last sentence of the discussion might be tempered accordingly.

We appreciate your comments. As you suggested, we would like to discuss about the clinical application and safety issue of our senolytic vaccine. In terms of application, CD153+ T-cells in visceral fat is increased with aging, and initial clinical application of this senolytic vaccine might be limited for obese and diabetes in old age. However, CD153+ T-cells might be possibly related to several age-related diseases as well as diabetes, we hope this technology will be able to be applied to other age-related diseases in future. In terms of safety issue, CD153+ T cells are reported as a mediator of CD4 T-cell-dependent control of Mycobacterium tuberculosis infection in the lung tissue (Nat Microbio 2018). In a clinical site, similarly, tumor necrosis factor alpha (TNF- α) inhibitors is associated with an increased risk of tuberculosis with the highest relative risks, 29.3 and 18.6, associated with adalimumab and infliximab, respectively. To avoid the serious side effect of TNF- α inhibitors, screening and treatment for latent tuberculosis infection (LTBI) in patients is therefore indicated. As a screening test, use of the tuberculin skin test (TST) in combination with Interferon gamma release assays (IGRAs) is justified to increase sensitivity. Commonly recommended LTBI treatment regimens in patients due to receive TNF- α inhibitors are daily isoniazid for 6 to 9 months. Indeed, LTBI screening programs prior to commencement of anti-TNF- α treatment significantly reduced the incidence of tuberculosis (Biologic Agents and Tuberculosis. Microbiol Spectr. 2016). Thus, the risk of CD153 vaccine for tuberculosis infection will be able to managed by following this protocol. Toward clinical application of CD153-CpG vaccine, the safety issue should be further discussed based on these clinical evidences. Accordingly, we mentioned about the risk of CD153-CpG vaccine in discussion (Page 8, Line 185), and changed the final sentence as follows, “Here, we propose that the CD153-CpG vaccine might be an optional tool for

senolytic therapy, and further safety evaluation and management will be required toward clinical application.” (Page 8, Line 189)

2. In addition, in Figure 4, panels C & D are expressed as percentage of crown-like structures around adipocytes and CD153+CLS. The data should be expressed as actual values. N = 3 mice per group. How many sections? How many fields counted? How many actual CLS observed? And how many stained positive for CD153? This is important to provide an indication as to what extent senescent T cells must be cleared to have a physiological impact.

We appreciate your comments. In Figure 4, we counted one section per sample and 40–50 fields in low power field ($\times 100$). According to your suggestion, percentage of F4/80⁺ crown-like structures (CLS) and CD153⁺ CLS around adipocytes has been changed into numbers of them in Figure 4C and 4D (Page 17, Line 424 and 425). The detail data has been shown in data set file. Consist with the previous analysis, immunohistochemical staining for F4/80 showed that fewer F4/80⁺ cells accumulated in the CLS of mice immunized with the CD153-CpG vaccine than in HFD control group mice (Fig. 4C). Immunohistochemical staining for CD153 also showed that the accumulation of CD153⁺ SA-T cells in CLS was ameliorated in the CD153-CpG-vaccinated group compared with that in HFD control group mice (Fig. 4D).

Response to Reviewer #2 (MD ID: NCOMMS-19-00198-T)

Thank you very much for your very important comments. According to your suggestions, we have certainly tried our best to revise the manuscript.

1. Mice vaccinated with CD-153-CpG vaccine described in the paper have decrease food consumption and body weight compared to mock vaccinated and high fat feed controls. Decrease food consumption in mice indicates stress or illness, unless intervention impacts CNS. As authors stated CD-153-CpG vaccinated mice after 11 weeks of high fed feeding have improved insulin glucose tolerance and insulin sensitivity and less CD153+SA T lymphocytes in the visceral fat. Many study endpoints described in the experiment with high fat feeding are sensitive to food intake, body weight and the level of adiposity, and it is important to clarify whether reported outcomes involving these measures are truly reflective of efficacy of vaccine or are a result of an adverse event/ toxicity of the immunization. To clarify these authors should perform “pair feeding”, and have additional control groups in the experiment. They need to vaccinate chow fed mice and look if there is any adverse or off target effect.

We appreciate your comments. Accordingly, we performed the same experiment with pair feeding and evaluated the effect of CD153-CpG vaccine on insulin resistance and glucose tolerance test. As a result, food intake and body weight among CD153-CpG vaccinated, KLH-CpG-vaccinated and HFD control groups were almost same under the condition of pair feeding (Supplement Fig.4). In this additional experiment, the glucose tolerance and insulin sensitivity were remarkably improved in the CD153-CpG-vaccinated group, compared with those in the KLH-CpG-vaccinated and HFD control groups (Supplement Fig.5). These results further indicated that CD153-CpG vaccination improves obesity-induced metabolic disorders, such as glucose tolerance and insulin resistance (Page 4, Line 97). In addition, we also administered the CD153-CpG vaccine to normal-diet (ND)-loaded mice. The body weight and food intake were almost same between the ND group and the ND (CD153-CpG) group (Supplement Fig.5), and the glucose tolerance and insulin sensitivity were also almost same (Supplement Fig.5). (Page 5, Line 101). On histological analysis of VAT, kidney tissues and lung tissues, no apparent differences were observed between ND group and ND (CD-153-CpG) group (Supplement Fig.11). These results indicated that CD153-CpG might not have severe adverse events such as weight loss and loss of

appetite.

2. Vaccination against CD153 positive lymphocytes might be risky and has very limited if any path to translation. CD153 expressing CD4+ lymphocytes play an important role in modulating immune response and fight infection (Nature Microbiology 2018 2018/11/01;3(11):1198-205). Vaccine which has prolonged sustainable effect against this ligand might have serious adverse effects on health. Author's should quote this paper and address this issue in the limitation of the approach.

There are publications showing beneficial effects of senescent cell clearance on metabolic health.

Authors should quote these publications to make the paper less superficial

Xu M, Palmer AK, Ding H, Weivoda MM, Pirtskhalava T, White TA, Sepe A, Johnson KO, Stout MB, Giorgadze N, Jensen MD, LeBrasseur NK, Tchkonina T, Kirkland JL. Targeting senescent cells enhances adipogenesis and metabolic function in old age. Elife 2015 Dec 19;4:e12997.

Schafer MJ, White TA, Evans G, Tonne JM, Verzosa GC, Stout MB, Mazula DL, Palmer AK, Baker DJ, Jensen MD, Torbenson MS, Miller JD, Ikeda Y, Tchkonina T, van Deursen JM, Kirkland JL, LeBrasseur NK. Exercise Prevents Diet-Induced Cellular Senescence in Adipose Tissue. Diabetes 2016 Jun;65(6):1606-15.

We appreciate your comments. In terms of safety issue, CD153 T cells are reported as a mediator of CD4 T-cell-dependent control of Mycobacterium tuberculosis infection in the lung tissue (Nat Microbio 2018). In a clinical site, tumor necrosis factor alpha (TNF- α) inhibitors is similarly associated with an increased risk of tuberculosis with the highest relative risks, 29.3 and 18.6, associated with adalimumab and infliximab, respectively. To avoid the serious side effect of TNF- α inhibitors, screening and treatment for latent tuberculosis infection (LTBI) in patients is therefore indicated. As a screening test, use of the tuberculin skin test (TST) in combination with Interferon gamma release assays (IGRAs) is justified to increase sensitivity. Commonly recommended LTBI treatment regimens in patients due to receive TNF- α inhibitors are daily isoniazid for 6 to 9 months. Indeed, LTBI screening programs prior to commencement of anti-TNF- α treatment significantly reduce the incidence of tuberculosis (Biologic Agents and Tuberculosis. Microbiol Spectr. 2016). Thus, the risk of CD153 vaccine for tuberculosis infection will be able to managed by following this protocol. Toward clinical application of CD153-CpG vaccine, the safety issue should be

further discussed based on these clinical evidences. Accordingly, we mentioned about the risk of CD153-CpG vaccine in discussion (Page 8, Line 185).

In addition, we referred your recommended study related to senescent cells in adipose tissue as follows, “Senescent cells accumulate in fat in aging, and exercise-mediated reduction as well as genetic clearance improved glucose metabolism or lipotoxicity, respectively” (Page 2, Line 22).

Response to Reviewer #3 (MD ID: NCOMMS-19-00198-T)

Thank you very much for your very important comments. According to your suggestions, we have certainly tried our best to revise the manuscript.

In the manuscript “The CD153 vaccine with CpG adjuvant is a novel serotherapeutic option for removing senescence-associated T cells from visceral adipose tissues”, the authors show that vaccination can delete cells that contribute to a senescent phenotype and improve obesity-induced metabolic disorders. I was asked to comment on the immunological aspects of this manuscript. The vaccination protocol and assessment of the response to vaccination are all appropriate and standard for the field. Quantification of antibody production and function (in a cytotoxic assay) are all done properly as is the flow cytometric analysis of cell populations. The only issue that I have is with the animals used in these studies. In the TLR7 ligand administration study, female mice were used, while in the HFD loading study, male mice were used. It is well known that males and females have different immune responses to vaccination. I would have liked to see both sexes be used for both experiments.

Aside from this point, this is a well done, nicely presented study.

I really appreciate your comments. According to your suggestion, the additional experiments are performed in this revised manuscript. In the TLR7 ligand administration study, eight-week-old male C57BL/6J mice as well as female C57BL/6N mice were utilized to evaluate the effect of CD153-CpG vaccine (Fig. 1E and Supple Fig. 2B). Although we utilized female C57 BL/6N mice in previous manuscript, we used same mice strain (male C57BL/6J mice) with high fat diet experiment in this revised manuscript.

The anti-CD153 antibody titer induced by the CD153-CpG vaccine maintained a high level during the administration of R848 (Supple Fig. 3A). Flow cytometry analysis showed that the proportion of splenic CD153⁺ SA-T cells was increased in the R848-treated mice at 18 weeks of age and increased to a lesser extent at 16 weeks of age compared with that in phosphate-buffered saline (PBS) control group mice. Notably, the proportion of splenic CD153⁺ SA-T cells was significantly decreased in mice immunized with the CD153-CpG vaccine compared with those in R848 control group mice and KLH-CpG-vaccinated mice (Fig. 1E and Supplement Fig. 3B). These results indicate that CD153-CpG vaccination decreases the number of splenic CD153⁺ SA-T

cells in both male C57BL/6J mice as well as female C57BL/6N mice, stimulated by chronic immune activation (page 4, Line 61).

In the high fat diet-induced diabetes study, male mice were recommended to evaluate the insulin resistance and glucose tolerance. Female mice usually have menstrual cycle and different estrogen levels in each mouse. Estrogen levels may affect the insulin sensitivity and glucose metabolism, and male mice are usually utilized for evaluation of glucose metabolism. Thus, in our experiments, male C57BL/6J mice were utilized to evaluate the effect of CD153-CpG vaccine on high fat diet-induced diabetes study.

Response to Reviewer #4 (MD ID: NCOMMS-19-00198-T)

Thank you very much for your very important comments. According to your suggestions, we have certainly tried our best to revise the manuscript.

1. Why did the authors chose CpG 1585 (Invivogen), a class A CpG, rather than a Class B or C CpG? It would be interesting to compare each of these classes of TLR9 agonists in your model.

We appreciate your comments. According to your suggestions, we used a class B or C CpG as an adjuvant which were compared with CpG 1585, a class A CpG in the TLR7 ligand administration study. Unexpectedly, co-treatment of a class B CpG with CD153 vaccine further increased anti-CD153 IgG2b and IgG2c antibody titer than that of a class A or C CpG adjuvant. However, there is no difference among these adjuvants treated mice in CDC assay. Although these additional experiments will give us the novel findings to improve our senolytic vaccine system, they are not main aim in our manuscript. Thus, these results are described in Supplement Figure only for Reviewer.

2. Is the mouse CD153#D peptide homologous in humans? In other words, could this approach be easily translated to vaccine development for humans?

We appreciate your comments. The 5 amino acids were different among sequence of mouse and human CD153#D. To examine the cross-reactivity with each other, human CD153#D peptide vaccine as well as mouse CD153#D was injected into mice and evaluate the antibody titer for both mouse and human CD153#D. Indeed, human CD153#D vaccine-induced antibody successfully cross-react with human, less than human, vice versa, mouse CD153#D vaccine-induced antibody cross-react with mouse, but not human. Thus, toward clinical application, human CD153#D peptide vaccine has to be utilized. The results are described in Discussion part and Supplement Figure 11.

3. Can you provide statistical analyses for Figure 1A and Figure 1C?

We appreciate your comments. According to your suggestions, we performed statistical analysis. In Figure 1A, we performed one-way ANOVA ($p < 0.0001$) among six group. The figure would be more complicated using Bonferroni's multiple comparison tests (D vs KLH, D vs A, D vs B, D vs C; $p < 0.0001$, D vs E; $p < 0.0018$). In Figure 1C, we added a new figure (Ratio to IgG1) and performed unpaired two-tailed t-test (natural

logarithmic transformation).

4. Why only 3 mice in Figure 1F?

I appreciate your comments. In this revised manuscript, the additional experiments are performed in the TLR7 ligand administration study. Eight-week-old male mice as well as female mice were utilized to evaluate the effect of CD153-CpG vaccine (Fig. 1E and Supple Fig. 3B). The results were shown as the proportion of SA-T cells induced by R848 administration in splenic tissues of male C57BL/6J mice (n = 6 in each group) at the ages of 18 weeks with or without the CD153-CpG vaccine or the KLH-CpG vaccine.

We corrected the manuscript as below lists.

P2, Line 22 “Senescent cells accumulate in fat in aging, and exercise-mediated reduction as well as genetic clearance improved glucose metabolism or lipotoxicity, respectively^{9,10}.” has been added.

P 7, Line 8 “Here, we propose that the CD153-CpG vaccine might be an optional tool for senolytic therapy” has been changed into “Here, we propose that the CD153-CpG vaccine might be an optional tool for senolytic therapy, and further safety evaluation will be required toward clinical application.”

P3, Line 60 “male C57Bl/6J mice” have been added.

P4. Line 97 “Furthermore, we performed pair-feeding among HFD-loaded mice and administered the CD153-CpG vaccine to normal-diet (ND)-loaded mice to clarify whether our reported outcomes on HFD loading study were truly reflective of the efficacy of CD153-CpG vaccination or were a result of an adverse event of the immunization. Under pair feeding conditions, the body weight of HFD-loaded mice had almost similar body weight (Supple Fig. 4A and 4B). The body weight and the weekly average amount of ND intake were almost similar between the ND control group and the ND group immunized with the CD153-CpG vaccine (Supple Fig. 4A and 4C). In the CD153-CpG-vaccinated group, despite the similar body weight and food intake, the insulin sensitivity (Supple Fig. 5A and 5B), the glucose tolerance (Supple Fig. 5C and 5D) and the HOMA-IR (Supple Fig. 5E) were improved compared with those in the KLH-CpG-vaccinated and HFD control groups. On the other hand, the insulin sensitivity, the glucose tolerance and the HOMA-IR were not significantly different between the ND control group and the ND group immunized with the CD153-CpG vaccine (Supple Fig. 5A–5E).” has been added.

P7, Line 169 “Furthermore, we also designed a human CD153#D peptide vaccine selected from human CD153 peptide (71–80 aa) homologous to mouse CD153#D peptide (76–85 aa) (Supple Fig. 2A) to examine the cross-reactivity with human and mouse. The human CD153#D-Alum vaccine was administered to seven-week-old male C57BL/6J mice three times at two-week intervals as well as mouse CD153#D-Alum vaccine, and human or mouse CD153 antibody production was evaluated by ELISA. We found the antibody induced by the human CD153#D vaccine strongly reacted with

human CD153#D-BSA and recombinant human CD153 (rhCD153), while it did not react with mouse CD153#D-BSA and rmCD153 (Supple Fig. 2B and 2C). The antibody induced by the mouse CD153#D vaccine hardly reacted with rhCD153 (Supple Fig. 2D), unlike with rmCD153.” has been added.

P8, Line 185 “CD153 T cells are reported as a mediator of CD4 T-cell-dependent control of Mycobacterium tuberculosis infection in the lung tissue²⁷. Although tumor necrosis factor alpha (TNF- α) inhibitors is associated with an increased risk of tuberculosis infection, screening and treatment for latent tuberculosis infection in patients is effective to reduce the incidence of tuberculosis²⁸. Toward clinical application of CD153-CpG vaccine, the safety evaluation and management should be further discussed based on these previous evidence. Here, we propose that the CD153-CpG vaccine might be an optional tool for senolytic therapy, and further safety evaluation and management will be required toward clinical application.” has been added.

Method section

P9. Line 211 “Mice had free access to food and water.” has been changed into “Mice had free access to food and water except for mice under pair-feeding condition.”

P9. Line 228 “male C57BL/6J mice” has been added.

P11. Line 286 “The serum concentration of insulin was determined using the Mouse Insulin ELISA kit (MercoDIA) according to the manufacturers“ has been added.

P12. Line 289 “and the results were analyzed using MMS-2 software (Muromachi Kikai)” has been added.

P12, Line 299 “VAT were collected from epididymal fat pads” has been added.

P12, Line 309 “BD FACS Diva software version 8.0.1” has been added.

P13, Line 339 “analysis of variance (ANOVA) test,” has been added.

Figure Legends

P16, Line 359 “Left figure; The titer against CD153-BSA is expressed as the dilution

of serum providing half-maximal binding (OD 50%). Right figure; Ratio of the titer against CD153-BSA-specific IgG2b:IgG1, IgG2c:IgG1 and IgG3:IgG1.” has been added.

P16, Line 424 and 425 “40–250 fields per sample,” has been added.

P14-16 The method of statistical analysis has been described in each figure.

References

P18, Line 446 New references were added as below.

9. Schafer, M. J. *et al.* Exercise Prevents Diet-Induced Cellular Senescence in Adipose Tissue. *Diabetes* **65**, 1606-1615 (2016).
10. Xu, M. *et al.* Targeting senescent cells enhances adipogenesis and metabolic function in old age. *eLife* **4**, 12997 (2015).

P19, Line 482 New references were added as below.

27. Sallin, M. A. *et al.* Host resistance to pulmonary Mycobacterium tuberculosis infection requires CD153 expression. *Nat. Microbiol.* **3**, 1198-1205 (2018).
28. Dobler, C. C. Biologic Agents and Tuberculosis. *Microbiol Spectr.* **4** (2016) doi: 10.1128/microbiolspec.TNMI7-0026-2016.

Reviewers' Comments:

Reviewer #1:

Remarks to the Author:

This is an interesting study that suggests that vaccination against CD153, a T cell marker, improves metabolic homeostasis in the face of a high fat diet. The manuscript is revised and the authors did a thorough job doing the requested additional experiments, including multiple additional in vivo studies.

My expertise is in senescence and senolytic therapies, not immunology. My concerns are:

- 1) What is the rationale for a vaccine as opposed to simply treating with an antibody against CD153?
- 2) The experimental design addresses the prevention of accumulation of CD153+ T cells during high fat feeding rather than their removal in for example an aged, obese, diabetic organism.
- 3) What is a senescent-associated T cell? Is it senescent itself? Or associated with senescent cells of non-immune lineages in tissues? SA-T is a very confusing nomenclature.
- 4) The main concern is that the title and manuscript are riddled with statements about senescence and apoptosis/cell death. These are both measurable entities. Yet neither are measured anywhere in this body of work. Thus, no statements about senotherapeutics or the impact of the vaccine on senescent cells can be made.

Reviewer #2:

Remarks to the Author:

All comments are addressed properly. in my opinion this is an interesting proof of the concept study, showing senotherapeutic effect of proposed vaccine against CD153 positive cells and will be interesting for readers.

Reviewer #4:

Remarks to the Author:

The authors have addressed all of my concerns.

Response to Reviewer #1 (MD ID: NCOMMS-19-00198-T)

Thank you very much for your very important comments. According to your suggestions, we have certainly tried our best to revise the manuscript.

1) What is the rationale for a vaccine as opposed to simply treating with an antibody against CD153?

I appreciate your comment. The therapeutic strategy of CD153 vaccine is almost same with that of CD153 antibody, and the long-term administration of CD153 neutralizing antibody could be also effective to prevent the accumulation of the senescent-associated T cells (SA-T cells). However, the high cost and frequent hospital visits for the antibody therapy are heavy burdens for the patients. Thus, the antibody therapy for long-term treatment may not be suitable for senolytic therapy. If the therapeutic vaccine has been developed as an alternative treatment of antibody drug, the patients will have the benefits to select the therapeutic option.

2) The experimental design addresses the prevention of accumulation of CD153+ T cells during high fat feeding rather than their removal in for example an aged, obese, diabetic organism.

I agree with your comment. According your suggestion, we revised our manuscript and changed the description from “to remove aging cells” to “to prevent the accumulation of aging cells”.

3) What is a senescent-associated T cell? Is it senescent itself? Or associated with senescent cells of non-immune lineages in tissues? SA-T is a very confusing nomenclature.

I appreciate your comment. According your suggestion, we performed the additional experiment to characterize the senescent-associated T cells (SA-T cells). In the previous literature, SA-T cells exhibited a marked increase in the expression of SA-cell cycle inhibitors (*Cdkn1a* and *Cdkn2b*) and SA-heterochromatin foci (SAHFs) (Fukushima Y, Minato N et al. *Inflamm Regen* 2018). Indeed, our results demonstrated that the majority of SA-T cells expressed senescence-associated β -galactosidase (SA- β -gal), a

typical cell senescence marker, and showed remarkably higher expression of γ -H2AX compared with the CD153⁻ counterpart cells, which is indicative of greater exposure to genostress. These results suggest that the SA-T cells show signatures of cell senescence. We added these results in the revised manuscript.

4) The main concern is that the title and manuscript are riddled with statements about senescence and apoptosis/cell death. These are both measurable entities. Yet neither are measured anywhere in this body of work. Thus, no statements about senotherapeutics or the impact of the vaccine on senescent cells can be made.

I appreciate your comment. According your suggestion, we added the analysis of apoptosis/cell death using TUNEL staining or senescence using immunostaining for p16^{ink4a} or γ -H2AX in adipose tissue. As a result, TUNEL-positive cells were markedly increased, and senescent cells such as p16^{ink4a}-positive cells and γ -H2AX-positive cells tended to be decreased in the VAT of CD153-CpG-vaccinated mice (Supple Fig. 11). It might be suggested that CD153-CpG vaccine induces apoptosis/cell-death and ameliorates the accumulation of senescent cells. We added these results and conclusion in the revised manuscript.

We corrected the manuscript as below:

P1. Title “The CD153 vaccine with CpG adjuvant is a novel serotherapeutic option for removing senescence-associated T cells from visceral adipose tissues” has been changed into “The CD153 vaccine with CpG adjuvant is a novel senotherapeutic option for preventing the accumulation of senescence-associated T cells from visceral adipose tissues”

P3. Line 55 The following sentences were added, “Consistent with a previous report¹¹, the majority of SA-T cells showed higher expression of senescence-associated β -galactosidase (SA- β -gal), a typical cell senescence marker, and phosphorylated histone H2AX at serine 139 (γ -H2AX), a DNA damage and repair marker, compared with the CD153⁻ counterpart cells, which is indicative of greater exposure to genostress. These results suggest that the SA-T cells show signatures of cell senescence (Supple Fig. 3).”

P6. Line 146 The following sentences were added, “Of importance, TUNEL-positive cells were markedly increased, and senescent cells such as p16^{ink4a}-positive cells and γ -H2AX-positive cells tended to be decreased in the VAT of CD153-CpG-vaccinated mice (Supple Fig. 11). It might be suggested that CD153-CpG vaccine induces apoptosis/cell-death and ameliorates the accumulation of senescent cells.”

P7, Line 173 “to remove aging cells” has been changed into “to prevent the accumulation of senescence cells by inducing apoptosis/cell death”.

P 7, Line 187 “to remove aging T cells” has been changed into “to prevent the accumulation of senescence T cells”.

P13. Line 318 The following sentences were added, “In addition, intracellular staining of phosphorylated histone H2AX at serine 139 (defined as γ -H2AX) was also performed using a fixation/permeabilization solution kit (BD Biosciences) according to the manufacturer’s recommendations. Blocking Fc-receptors and staining cells with a mixture of fluorescently labeled antibodies, CD4-APC-Cy7 (RM4-5, BD Biosciences), CD44-PE, CD153-BV421 and PD-1-APC, was performed before intracellular staining of γ -H2AX-Alexa Fluor 488 (2F3, Bio Legend).”

P13. Line 326 The following sentences were added, “SA- β -gal assay was performed using a cellular senescence detection kit (SPiDER- β Gal, Dojindo).”

P14. Immunohistochemistry The description of immunohistochemical staining for p16^{ink4a} or γ -H2AX and TUNEL staining were added.

Supplement Figures

Supplement Figure 3 and 11 were added in this revised manuscript. Thus, the number of supplement figures were accordingly changed.

Reviewers' Comments:

Reviewer #1:

The authors have been quite responsive to my comments. However, I still have some reservations as outlined below. You are welcome to transmit my reservations to the authors directly or with modification.

1) The definition of senescence-associated T cells remains poorly defined. "Senescence-associated" implies associated with senescent cells of a different origin (than T cells). It appears that in fact the T cells themselves could be senescent. Thus, the descriptor "senescent associated" is misleading. Nevertheless, the phrase is in the published literature, whether accurate or not, so the authors should be allowed to use it, regardless of its accuracy (although I am hesitant to propagate inaccuracies).

2) There is still something jarring about the paper. Paragraph 1 talks about the value of targeting senescent cells therapeutically. Then paragraph 2 jumps in to describe a vaccination approach to killing senescent cells. Senoptosis/senolysis as an approach to targeting age-related decline is still in debate. Thus, it would be valuable to justify vaccination vs. other therapeutic approaches like small molecules or antibodies.

3) To determine if senescence-associated T cells are in fact senescent, the authors measured p16 and gH2AX in WAT section by IHC. While the representative images look reasonable, the quantitation of p16 signal does not reach significance (Sup Fig 11). This is most likely because it is well known (although not clearly published) that none of the commercially available antibodies against p16 work in mouse tissue. There is a signal on WB or IHC, but the same signal occurs in analysis of tissues from p16KO mice. I do not recommend including this data for that reason. qPCR to measure p16 expression is much preferred.

So while the study is innovative, addresses an important topic, demonstrates efficacy in terms of metabolic outcomes, it remains unclear in my mind what the vaccine is targeting precisely and the mechanism by which it works

Response to Reviewer #1 (MD ID: NCOMMS-19-00198-B)

Thank you very much for your very important comments. According to your suggestions, we have certainly tried our best to revise the manuscript.

1) The definition of senescence-associated T cells remains poorly defined. "Senescence-associated" implies associated with senescent cells of a different origin (than T cells). It appears that in fact the T cells themselves could be senescent. Thus, the descriptor "senescent associated" is misleading. Nevertheless, the phrase is in the published literature, whether accurate or not, so the authors should be allowed to use it, regardless of its accuracy (although I am hesitant to propagate inaccuracies).

I appreciate your comment. Our additional experiments and previous findings indicate that senescence-associated T cells show signatures of cell senescence. We carefully used this phrase in this manuscript, and a part of description was changed into senescence T cells.

We also mentioned about the study limitation to evaluate the cellular senescence, which was assessed by only higher expression of SA- β -gal and γ -H2AX) for SA-T cells. We added the below sentence in a Discussion section.

“Other study limitation is the evaluation method of cellular senescence assessed by only higher expression of SA- β -gal and γ -H2AX) for SA-T cells. The SA-T cells should be further discussed about their potency as senescent cells and contribution to several diseases.”

2) There is still something jarring about the paper. Paragraph 1 talks about the value of targeting senescent cells therapeutically. Then paragraph 2 jumps in to describe a vaccination approach to killing senescent cells. Senoptosis/senolysis as an approach to targeting age-related decline is still in debate. Thus, it would be valuable to justify vaccination vs. other therapeutic approaches like small molecules or antibodies.

I agree with your comment. According your suggestion, we revised our manuscript and added the introduction of senolysis in the paragraph 1.

3) To determine if senescence-associated T cells are in fact senescent, the authors

measured p16 and gH2AX in WAT section by IHC. While the representative images look reasonable, the quantitation of p16 signal does not reach significance (Sup Fig 11). This is most likely because it is well known (although not clearly published) that none of the commercially available antibodies against p16 work in mouse tissue. There is a signal on WB or IHC, but the same signal occurs in analysis of tissues from p16KO mice. I do not recommend including this data for that reason. qPCR to measure p16 expression is much preferred.

I appreciate your comment. We did not notice the reliability of the commercially available antibodies against p16 in mouse tissue. According your suggestion, we deleted the result of IHC against p16.